# Oxygen Saturation in Hospitalized COVID-19 Patients and Its Relation to Colchicine Treatment: A Retrospective Cohort Study with an Updated Systematic Review

**DOI:** 10.3390/medicina59050934

**Published:** 2023-05-12

**Authors:** Sandy Sharaf, Rasha Ashmawy, Eman Saleh, Mayada Salama, Yousra A. El-Maradny, Ali Zari, Shahinda Aly, Ahmed Tolba, Doaa Mahrous, Hanan Elsayed, Dalia Latif, Elrashdy M. Redwan, Ehab Kamal

**Affiliations:** 1Clinical Research Department, Maamora Chest Hospital, MoHP, Alexandria 21923, Egypt; Sandy.Mohsen@alexu.edu.eg (S.S.); mri.rasha.m.informatics19@alexu.edu.eg (R.A.); shahinda_elhamshary88@yahoo.com (S.A.); mahrousdoaa2@gmail.com (D.M.); 2Infectious Diseases Administration, Directorate of Health Affairs, MoHP, Alexandria 21554, Egypt; 3Clinical Research Department, El-Gomhoria General Hospital, MoHP, Alexandria 21566, Egypt; gs-eman.ibrahim@alexu.edu.eg (E.S.); mayadasalama@yahoo.com (M.S.); 4Microbiology and Immunology, Faculty of Pharmacy, Arab Academy for Science, Technology and Maritime Transport (AASTMT), Alamein 51718, Egypt; hiph.ymaradny@alexu.edu.eg; 5Protein Research Department, Genetic Engineering and Biotechnology Research Institute, City of Scientific Research and Technological Applications (SRTA-City), New Borg EL-Arab, Alexandria 21934, Egypt; 6Department of Biological Science, Faculty of Science, King Abdulaziz University, Jeddah 21589, Saudi Arabia; azari@kau.edu.sa; 7Princess Dr. Najlaa Bint Saud Al-Saud Center for Excellence Research in Biotechnology, King Abdulaziz University, Jeddah 21589, Saudi Arabia; 8Clinical Research Department, Abou-Kir General Hospital, MoHP, Alexandria 21913, Egypt; phahmedtolba92@gmail.com (A.T.); dalia.mo.latif.99@gmail.com (D.L.); 9Department of Biomedical Informatics and Medical Statistics, Medical Research Institute, Alexandria University, Alexandria 21561, Egypt; mri.hanan.m.informatics18@alexu.edu.eg; 10Medical Research Division, National Research Center, Giza 12622, Egypt; ehabkamal2011@hotmail.com

**Keywords:** SARS-CoV-2, COVID-19, colchicine, oxygen supply, oxygen equipment, LOS, mortality, systematic review

## Abstract

*Background*: Colchicine has been proposed as a cytokine storm-blocking agent for COVID-19 due to its efficacy as an anti-inflammatory drug. The findings of the studies were contentious on the role of colchicine in preventing deterioration in COVID-19 patients. We aimed to evaluate the efficacy of colchicine in COVID-19-hospitalized patients. *Design*: A retrospective observational cohort study was carried out at three major isolation hospitals in Alexandria (Egypt), covering multiple centers. In addition, a systematic review was conducted by searching six different databases for published studies on the utilization of colchicine in patients with COVID-19 until March 2023. The primary outcome measure was to determine whether colchicine could decrease the number of days that the patient needed supplemental oxygen. The secondary outcomes were to evaluate whether colchicine could reduce the number of hospitalization days and mortality rate in these patients. *Results*: Out of 515 hospitalized COVID-19 patients, 411 were included in the survival analysis. After adjusting for the patients’ characteristics, patients not receiving colchicine had a shorter length of stay (median: 7.0 vs. 6.0 days) and fewer days of supplemental oxygen treatment (median: 6.0 vs. 5.0 days), *p* < 0.05, but there was no significant difference in mortality rate. In a subgroup analysis based on oxygen equipment at admission, patients admitted on nasal cannula/face masks who did not receive colchicine had a shorter duration on oxygen supply than those who did [Hazard Ratio (HR) = 0.76 (CI 0.59–0.97)]. Using cox-regression analysis, clarithromycin compared to azithromycin in colchicine-treated patients was associated with a higher risk of longer duration on oxygen supply [HR = 1.77 (CI 1.04–2.99)]. Furthermore, we summarized 36 published colchicine studies, including 114,878 COVID-19 patients. *Conclusions:* COVID-19-hospitalized patients who were given colchicine had poorer outcomes in terms of the duration of supplemental oxygen use and the length of their hospital stay. Therefore, based on these findings, the use of colchicine is not recommended for COVID-19-hospitalized adults.

## 1. Introduction

Coronavirus 2019 (COVID-19) infection poses a threat to worldwide public health [1]; until 30 March 2023, 761.4 million cases and 6.9 million fatalities have been reported, and high mortality is associated with acute respiratory distress syndrome (ARDS) [2]. It is essential to investigate affordable treatments [3]. Despite licensing different types of COVID-19 vaccines, phase 4 trials are still ongoing, and COVID-19 patients still require therapeutic management against the virus and its complications.

The COVID-19 viral infection progresses through three phases: the early infection phase, the pulmonary phase, and the inflammatory phase. The deterioration and development of complications (ARDS, cardiovascular derangement, including endothelial damage and thrombosis, multi-organ (system) dysfunction, and fibrotic tissue that resulted in pulmonary fibrosis) started from the pulmonary phase through the production of pro-inflammatory markers (innate immunity) to a hyperactive systemic inflammatory state (cytokine storm) (adaptive immunity) [4,5]. This hyper-inflammatory immune response generates intracellular signaling cascades, and the excessive internal stimulation leads to cell death managed by the Nod-like receptor family, pyrin-containing domain 3 (NLRP3), through inflammasome activation [6]. These occurrences cause a cytokine storm (CS).

Consequently, the pulmonary phase is the crucial time to control COVID-19 infection, as it is a time for therapy by suppressing inflammation without aggressively suppressing immunity. Other anti-inflammatory drugs and immunomodulatory agents (corticosteroids, anti-interleukin-6 agents such as tocilizumab and sarilumab, and IL-1 receptor antagonists such as anakinra or canakinumab) have been tested in observational and randomized clinical trials. Currently, only dexamethasone and tocilizumab have achieved scientific evidence to improve the prognosis of patients with severe COVID-19 [7].

Colchicine is an old but effective anti-inflammatory medication that is commonly used to treat various conditions such as gout, pericarditis, and auto-inflammatory syndromes, including Bechet’s disease, adult-onset Still’s disease, and familial Mediterranean fever [8]. The medication has a multi-target mechanism of action, which has shown potential efficacy in managing the hyperactive inflammatory state of COVID-19 infection by targeting the cytokine cascade at different levels [9]. Unlike other anti-inflammatory drugs, colchicine does not have an immunosuppressive effect, and it inhibits the innate immune response without affecting the adaptive immune system. This feature helps prevent the complications mentioned earlier and acts as a cytokine storm-blocking agent [10]. In addition, colchicine is a lipid-soluble medication that reaches its peak plasma concentrations 1 hour after administration, with maximal anti-inflammatory effects occurring over 24 to 48 h due to its accumulation within leukocytes rather than plasma [11]. It is an affordable medication with a high tolerance and is considered safe according to established therapeutic guidelines, given its modest dose. Therefore, it was recommended to start colchicine during the first phase of the infection and continue throughout subsequent phases to help manage the disease effectively [12].

Numerous articles have investigated the potential of colchicine to prevent the deterioration of COVID-19 patients. In May 2021, Golpour et al. reported a reduced mortality rate and hospitalization duration in COVID-19 patients in a meta-analysis [13]. Another randomized clinical trial reported that colchicine shortened the period of supplementary oxygen therapy and hospitalization in hospitalized COVID-19 patients [14]. However, other studies have found no significance in colchicine for different outcomes; for instance, a May 2022 meta-analysis by Toro-Huamanchumo et al. reported no reduction in mortality or hospitalization duration in hospitalized COVID-19 patients who were given colchicine [15]. The RECOVERY trial, a large, randomized control trial, also reported similar findings to Mehta et al. in March 2022, concluding that colchicine did not reduce mortality, the need for ventilatory support and ICU admission, or the length of hospital stay in COVID-19 patients [16,17].

The objective of the present study was to investigate the potential role of colchicine in hospitalized COVID-19 patients and determine its efficacy as compared to standard care. Specifically, the study had three aims: firstly, to evaluate the effectiveness of colchicine initiation in the COVID-19 protocol in reducing the duration of treatment with supplemental oxygen while accounting for possible confounding factors using a Cox-hazard model. Secondly, to assess the potential benefit of using colchicine in reducing the length of hospital stay and mortality rate. Finally, to systematically review all published evidence regarding the use of colchicine in hospitalized COVID-19 patients.

## 2. Research Design and Methodology

A retrospective observational cohort study was carried out in three major isolation hospitals in Alexandria (Egypt), involving confirmed hospitalized COVID-19 patients (either critically or non-critically ill) who were divided into two groups: a colchicine group and a non-colchicine group.

### 2.1. Study Population

#### 2.1.1. Inclusion Criteria

All adult patients (aged more than 18 years old) admitted to hospital (wards, intermediate care units, or intensive care units) were diagnosed with COVID-19 through polymerase chain reaction (PCR) by using a deep nasopharyngeal swab from 1 June until the end of November 2021, either administering colchicine tablets or not, with any degree of severity according to the Egyptian MoHP protocol version 4, September 2021 [18], Appendix A and Appendix A.

#### 2.1.2. Exclusion Criteria

Patients who were receiving colchicine for a different medical condition, patients who were admitted for only one day (due to death or transfer), and patients with incomplete medical records were excluded from the study.

### 2.2. Study Drug (Exposure)

Colchicine was administered orally at 0.5 mg twice daily from admission day without loading, but it could be modified to 0.5 mg daily for renal impairment patients.

### 2.3. Data Collection

All possible patient data were collected from hospital medical records, by a trained research team, such as demographics, vital signs, comorbidities, and baseline laboratory tests such as complete blood count (CBC) by flow cytometry, C-reactive protein (CRP) by latex agglutination, random blood sugar (RBS) by glucose oxidase/hydrogen peroxide, urea by urease/ultraviolet (UV), serum creatinine (sCr) by alkaline picrate at 505 nm primary, 578 nm secondary respectively, and alanine aminotransferase (ALT), as well as aspartate aminotransferase (AST) by UV-spectrometry without addition of pyridoxal-5-phosphate (P-5-P) at 340 nm for and 405 nm for secondary, according to International Federation of Clinical Chemistry and Laboratory Medicine (IFCC). Furthermore, oxygen saturation in room air, mode of respiratory support needed (nasal oxygen, oxygen mask, mask reservoir, non-invasive, or invasive), and medications. All these data were collected by senior members of the medical team from the hospital’s medical records during a pre-specified period and recorded in an electronic data collection form. 

#### 2.3.1. The Primary Outcome

To determine whether the use of colchicine plus standard care therapy affects oxygen saturation by comparing the median days of supplemental oxygen treatment for both groups. 

#### 2.3.2. The Secondary Outcomes

The hospital length of stay (LOS) and mortality rate.

### 2.4. Statistical Analysis

First, descriptive analysis and appropriate tests based on normality. For continuous variables, if they followed a normal distribution, the mean and standard deviation (SD) were used; otherwise, the median and interquartile range (IQR) were used. Tests of significance between the two groups were either a *t*-test or Mann-Whitney. Categorical variables were shown as frequencies and percentages using the chi-squared test. Then, we performed Kaplan-Meier survival analysis and the log-rank test to compare the median days of supplemental oxygen treatment between the two groups. The patients were classified based on the type of supplemental oxygen they received upon admission (day 1), and Cox-hazard regression was used to adjust for any confounding factors. The significance level was set to less than 0.05 for all tests conducted. All data manipulation was performed using R version 4.1.1 (R Foundation for Statistical Computing, Vienna, Austria).

#### Sample Size Calculation

The calculation of the sample size was carried out using STATA 16 software (StataCorp. 2019. Stata Statistical Software: Release 16. College Station, TX: StataCorp LLC). Assuming mean oxygen saturation at the first 3-day interval in the standard care group (87.3 ± 10.4) and in the colchicine group (90.2 ± 4.9) (20), power is 90% at a 95% level of confidence. Therefore, the minimum accepted sample size is 166, rounded to 170 for each group (340 total).

### 2.5. Systematic Review

#### 2.5.1. Database Search and Study Selection

S.S., H.E., A.T., D.L., and Y.A.E.-M. searched independently 6 different databases [Scopus, PubMed, Clinical Trial.gov, Research Square, Research Gate, and medRixv] to retrieve all published evidence for using colchicine in COVID-19 patients until March 2023, using keywords (Coronavirus, Coronavirus infections, COVID 2019, SARS2, SARS-CoV-2, SARS-CoV-19, severe acute respiratory syndrome coronavirus 2, coronavirus infection, severe acute respiratory pneumonia outbreak, novel CoV, 2019 ncov, sars cov2, cov2, ncov, COVID-19, COVID19, coronavirus AND Colchicine) filtered by 2020, as shown in Appendix A. Furthermore, S.S., R.A., Y.A.E.-M., and E.K. were the second reviewers.

#### 2.5.2. Screening and Data Extraction

The resulting articles were stored at the Mendeley reference manager and used to check for duplicates. A.T. and D.L. screened the remaining articles independently by title and abstract. Subsequently, D.M. and H.E. read full articles to check for the eligibility of included studies for qualitative analysis. Where R.A., S.S., and Y.A.E.-M. were the second reviewers for all processes. Moreover, they extracted the data for qualitative analysis.

## 3. Results

This study included 515 COVID-19 patients who were hospitalized and confirmed by PCR. Among them, 259 were in the colchicine group and 256 were in the non-colchicine group. The average age of all patients was 60.9 ± 13.2, and patients in the colchicine group were older than those in the non-colchicine group (*p* = 0.041). The median oxygen saturation at admission was approximately 88% for both groups. Most patients were on an oxygen mask/nasal cannula on day one of admission (59.8%), while the fewest number of patients were on mechanical ventilation (3.1%). There was no significant difference between the two groups regarding oxygen saturation in room air (*p* = 0.327) or oxygen equipment at admission (*p* = 0.281). However, the respiratory rate was significantly different between the two groups (*p* < 0.0001), as shown in Table 1.

### 3.1. Outcomes

Surprisingly, patients not receiving colchicine had a lower length of stay and supplemental oxygen treatment days, and the median time was 6 compared to 5 days in the colchicine group (*p* = 0.021, 0.008), while there was no statistically significant difference in mortality rate (*p* = 0.468) (Table 2).

After performing a survival analysis on 411 patients who were discharged alive, 104 patients died to compare the in-hospital days of supplemental oxygen treatment. Almost half of the patients receiving colchicine stopped using supplemental oxygen for the sixth day (median 6.0; IQR 5.0–6.0 days) of intervention, while the same happened to the patients receiving placebo on the fifth day (median 5; IQR 5.0–6.0 days; *p* < 0.005). Upon subgroup analysis by the type of supplemental oxygen used on admission—nasal cannula/ oxygen mask, reservoir mask, or mechanical ventilation (MV), either invasive or noninvasive—we found that non-colchicine patients with a nasal cannula/oxygen mask class had fewer days of treatment with supplemental oxygen (median 5 vs. 6; *p* < 0.003). On the other hand, there was no significant difference for the mask reservoir subgroup. Unfortunately, we could not compare MV patients due to the high mortality in this class for both groups, as shown in Figure 1.

### 3.2. Adjusting Variables

Comorbidities and concomitant treatments were adjusted using Cox regression for patients admitted to the nasal cannula/face mask class. After adjusting for confounders (age, oxygen saturation on admission, gender, consciousness level at admission, macrolides, and use of corticosteroids), patients receiving colchicine had 1.24 times the risk of obtaining a longer duration of supplemental oxygen treatment days than the non-colchicine group [HR = 0.76 (CI 0.59–0.97)]. The only significant predictor was using clarithromycin in comparison to azithromycin [HR = 1.77 (CI 1.04–2.99)], which means that using clarithromycin increases in-hospital days of supplemental oxygen needed more than azithromycin. There were no differences between the two groups in comorbidities, but there was a statistically significant difference in macrolide use between the two groups (*p* = 0.001). The majority of both groups were on corticosteroid therapy (94.2%) and anticoagulant therapy (95.9%). Most patients were on LMWHs. Colchicine duration treatment median time, IQR, was 5 (3–7) days.

### 3.3. Systematic Review

Figure 2 shows that from a total of 957 articles, Mendeley detected 210 duplicates: 3 were incomplete studies (only published protocols), 708 were irrelevant and manually detected duplicates, and 36 eligible articles were included for qualitative analysis (35 full texts and one published abstract). A total of 23 studies were randomized controlled trials, thirteen were observational studies, and the total number of COVID-19 patients enrolled in 36 different studies was 114,878 adults (18–84 years), 25,866 from clinical trials, and 89,012 from observational studies. The quality assessment of the studies was performed using the Cochrane risk of bias tool for randomized controlled trials and adapted Newcastle–Ottawa scales for observational studies by total score as low, medium, and high risk of bias. After that, the Agency for Healthcare Research and Quality (AHRQ) reported that both were good, fair, or poor [19,20].

Studies varied in population, intervention, outcome, and risk of bias. Hospitalized COVID-19 patients were the target population in all studies; 3 studies mentioned that they targeted non-ICU/non-ventilated patients; 4 studies targeted severe patients; 2 studies targeted moderate to severe patients; 2 studies targeted mild patients; and 2 studies targeted only moderate patients. Only 16.7% of the studies compare colchicine to a placebo; others use colchicine as an add-on therapy, as shown in Table 3. The dose of colchicine and total duration were highly heterogeneous, as we could not categorize the studies.

Regarding outcomes, 23 studies assessed mortality (11 studies showed no effect on mortality); 14 studies assessed the length of hospital stay (5 studies showed no difference in LOS days); 29 studies assessed progression or improvement in the aspect of symptoms, oxygen supply, oxygen saturation, or level of hospitalization (17 studies revealed no difference); 6 studies assessed inflammation (50–50% on both sides).

RCTs included two studies with good quality assessment, five with fair quality assessment, and sixteen with poor quality assessment. Observational studies included one, six, and six studies with high, medium, and low risk of bias, respectively, and after being converted to AHRQ standards, there were eleven studies with good quality and two studies with poor quality.

## 4. Discussion

### 4.1. Based on Our Study Analysis

To the best of our knowledge, this is the first study to investigate the negative impact of administering colchicine as a treatment for hospitalized COVID-19 patients in terms of days of using supplemental oxygen. Other studies did not observe any deterioration in oxygen levels with colchicine and showed no significant differences compared to the control group. Our study found that patients who received colchicine had a longer duration of oxygen supplementation (median of 6.0 days versus 5.0 days) and a longer hospital stay (median of 7.0 days versus 6.0 days) than those who did not receive colchicine. Furthermore, this study was adequately powered to identify differences across specified sub-analyses concerning the oxygen equipment used upon admission. Specifically, we found that patients who were admitted using a nasal cannula/face mask (which constituted the majority of our patients) and did not receive colchicine had a shorter duration of oxygen supply than those who received colchicine [HR = 0.76; CI (0.59–0.97)].

Furthermore, the use of macrolides, specifically clarithromycin, compared to azithromycin in patients who received colchicine was associated with a higher risk of longer duration on oxygen supply [HR = 1.77; CI (1.04–2.99)] after adjustment by patients’ characteristics, comorbidities, and concurrent therapies (antivirals, anticoagulants, antibiotics, and others) by cox-regression analysis. Our findings were compelling in patients with low oxygen levels, but we were unable to evaluate medications’ effects in more severe cases since most of them died. Moreover, the presence of more risk in those taking clarithromycin with colchicine explains the observed hazardous effect; the interaction existed upon the co-administration of colchicine with cytochrome P450 inhibitors such as macrolides (erythromycin and clarithromycin), and clarithromycin has more potency for this receptor, which explains that increasing the dose of colchicine with these inhibitors, in turn, increases the risk [55]. Furthermore, Kamel’s review supported the idea that combining antivirals and macrolides with colchicine treatment for COVID-19 patients could reduce its positive anti-inflammatory impact [55].

Although there was a higher mortality rate in patients without colchicine treatment (21.5% vs. 18.9%), there was no statistically significant difference between the two groups.

### 4.2. Based on Our Systematic Review Analysis

We have determined that over 50% of the studies observed no statistically significant reduction in mortality, symptom limitation, hospitalization, oxygen supply requirements, or inflammation reduction. The majority of these studies were randomized controlled trials (RCTs), and the highest-quality study with a large sample size confirmed this conclusion [17,25,30,32,33], except for one study that had a small sample size and mainly included obese patients among the 72 participants, thus limiting the generalizability of its findings [14]. However, it is worth noting that most studies supporting the use of colchicine were conducted before the emergence of the Delta and Omicron variants of concern (VOCs), which may have influenced the severity of cases. Moreover, the spread of VOCs varied across different countries, which could also have impacted the results given that these studies were conducted in various locations around the world. In addition, not all studies took into account the effects of COVID-19 vaccinations, which began towards the end of 2020 and may have improved the outcomes in the control group. Other variables that may have influenced the results include variations in the colchicine dosage and the presence or absence of loading doses, the standard of care (SOC) with or without antivirals and corticosteroids, and the severity and age range of cases, among others. Lastly, in the observational studies, the only study with a large sample size found no significant association between the use of colchicine and the risk of hospitalization or disease severity [54].

Two RCTs have the same primary endpoint (oxygen supply days), one with a low sample size and the other with variable results between different ages, both with poor quality assessments [28,36]. Although our study had a larger sample size (n = 411) and a long study period, it should be noted that. Similar to our findings, the major randomized RECOVERY study’s 6-day median length of colchicine therapy (IQR 3–9 days) revealed no differences in death, time to discharge, or progression to critical disease [17]. In addition, we considered including the lowest dose without a loading dose compared to prior trials, and the majority of our patients had normal renal and hepatic function. Concerning our primary outcome, the NCT04324463 (ACT, an open-label, factorial, randomized, controlled trial) concluded no difference in high oxygen flow, mechanical ventilation, death, or respiratory death and suggests that colchicine should not be used for the treatment of hospitalized COVID-19 patients [35]. Additionally, Perricone et al.’s multi-center, open-labeled clinical trial testing a high dose of colchicine for a long duration reported no difference in oxygen requirements, days of hospitalization, or comorbidities [38]. Nevertheless, even colchicine use in the early stages did not modify the risk of hospitalization, prevention, or progression of the disease [54]. Additionally, a randomized controlled trial (RCT) also supported our findings, which led to the early termination of the study. The use of a combination of colchicine and rosuvastatin was found to have a negative impact on non-critically hospitalized COVID-19 patients as it failed to prevent disease progression in multiple aspects, including the need for oxygen supplementation [56].

Other studies that demonstrate the effectiveness of colchicine therapy in hospitalized COVID-19 patients using concurrent medicines had small sample sizes, were carried out in a single center, or were limited to some medications [42,44]. Other small-scale studies that did not detect a significant difference in the outcomes either excluded patients who received azithromycin and antimalarial medications for COVID-19 or analyzed just dexamethasone and remdesivir as adjuvant medications [15,22].

To examine the effectiveness and safety of colchicine in a range of COVID-19 settings, several published meta-analyses have been carried out. The findings of Elshafei et al., Golpour et al., Beran et al., and Salah et al. are consistent with each other since they all demonstrated a beneficial effect of colchicine use [11,13,57,58]. However, there are several issues with the analysis of these studies, as they employed both randomized and observational methods. Combining data from studies with different designs into one meta-analysis can result in skewed outcomes and inaccurate conclusions, which is not practical in reality. On the other hand, large meta-analyses that only analyzed randomized studies have found no benefits at the endpoints [15,16].

Studies were not able to justify the withdrawal of colchicine use; our study might be a step forward to evaluate the efficacy of colchicine in COVID-19-hospitalized patients. We are unable to explain the mechanism by which colchicine prolongs the duration of the oxygen supply. However, the colchicine group had a significantly higher number of pulmonary embolism (PE) cases in the largest study performed by Tardiff et al., and the trial was stopped before the scheduled sample size had been fully enrolled due to logistical reasons [59]. Additionally, it has been previously shown that colchicine decreases the production of surfactants at high therapeutic dosages, which may raise the risk of ARDS and multi-organ failure in COVID-19. Additionally, we suggest that this may be attributed to the side effects of colchicine (myopathy and bone marrow suppression), especially with corticosteroid therapy and in elderly patients. To identify the hazards of colchicine in the treatment of COVID-19 patients, more powerful studies and pooled analyses are required. The molecular pathophysiology and mechanism behind COVID-19 infection are currently incomplete and unclear. Thus, unanticipated consequences might be disregarded in the rush to create COVID-19 treatment options [60].

### 4.3. Strengths and Limitations

Our study has several strengths. First, it is the first large-scale, multi-center retrospective cohort study to reveal that colchicine has a negative impact on the treatment of COVID-19. Additionally, we were able to detect this impact in less severe individuals without excluding any patients with concurrent therapies. Second, we conducted a recent and comprehensive systematic search of six databases. Third, we do sub-survival analysis using oxygen-supplemental equipment. Fourth, we adjusted for all potential confounders, which strengthened our conclusion.

However, there are certain limitations: (1) The loss of follow-up of patients prohibits us from detecting adverse events, but we could detect the risk with other medications that have defects in other studies. (2) It is retrospective in nature and lacks some detailed information such as BMI and laboratory markers (ferritin, IL, D-dimer, and CRP) in most patients. (3) The effect of colchicine on inflammatory biomarkers cannot be assessed. However, while trials showed an effect on biomarkers, they were ambiguous on the impact on critical outcomes, suggesting that colchicine is not potent enough to counteract the cytokine storm [21,22]. Worthy of note is the fact that most patients receiving corticosteroids at the same time as colchicine might have influenced the measure of differences over the control group. So, further trials would be necessary to clarify the effect on inflammatory biomarkers in early COVID-19 patients without other anti-inflammatory drugs. Currently, guidelines for the management of hospitalized adults with COVID-19 recommend against the use of colchicine. Our study contributes to the reinforcement of this recommendation and will be useful for front-line physicians facing COVID-19 patients.

## 5. Conclusions

In this retrospective cohort study of substantial size, it was observed that patients who were administered colchicine displayed the worst outcomes concerning their need for supplemental oxygen and length of hospitalization. Therefore, the use of colchicine in hospitalized adults with COVID-19 is not recommended. Additionally, the causes of discrepancies in the outcomes of prior studies are not fully comprehended and may be attributed to several factors, such as dissimilar COVID-19 treatment protocols, the timing of colchicine administration, immunity status against the virus, patients’ disease severity, patients’ comorbidities or not, and variations in SARS-CoV-2 variants of concern.

## Figures and Tables

**Figure 1 medicina-59-00934-f001:**
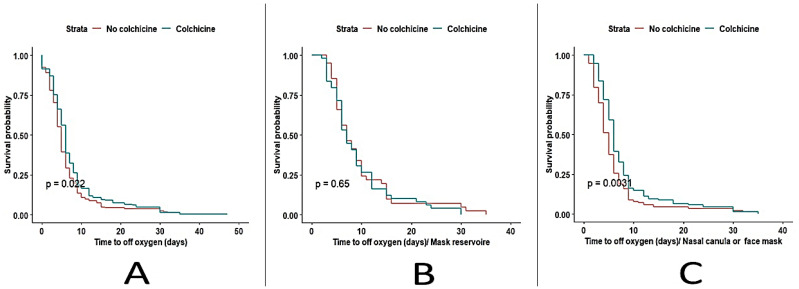
Kaplan–Meier curves of time to the end of need for supplemental oxygen for both groups (**A**), oxygen supplement/mask reservoir group (**B**), and oxygen supplement/ nasal canula or face mask group (**C**).

**Figure 2 medicina-59-00934-f002:**
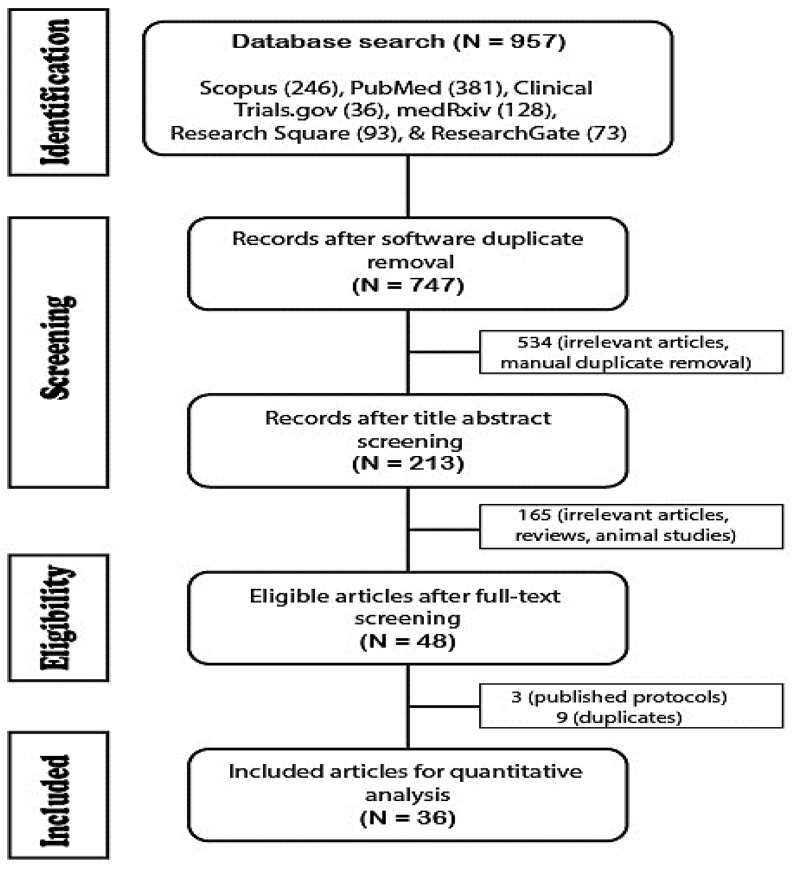
PRISMA flow chart showing screened and included articles.

**Table 1 medicina-59-00934-t001:** Baseline characteristics of the patients.

Characteristics	Colchicine Group (N = 259)	Non-Colchicine Group (N = 256)	Total (N = 515)	*p*-Value
**Gender** (Female), N (%)	167 (64.5%)	164 (64.1%)	331 (64.3%)	0.921
**Age (years),** Mean ± SD	62.0 ± 12.4	59.8 ± 13.9	60.9 ± 13.2	0.061
**Age category** (≥50), N (%)	216 (83.4%)	195 (76.2%)	411 (79.8%)	0.041
**Level of consciousness** (Alert), N (%)	235 (90.7%)	222 (86.7%)	457 (88.7%)	0.15
**Heart rate (Beat/minute)**, Median (IQR)	87 (82, 90)	88 (81, 90)	87 (82, 90)	0.311
**Respiratory rate (cycle /minute)**, Median (IQR))	22 (21, 23)	23 (22, 26)	22 (22, 24.5)	<0.001
**O_2_ saturation%**, Median (IQR)	88 (80, 91)	88.0 (80, 92)	88 (80, 91)	0.327
**Blood pressure**, N (%)				0.601
Hypertension	36 (13.9%)	30 (11.7%)	66 (12.8%)	
Hypotension	1 (0.4%)	2 (0.8%)	3 (0.6%)	
Normal	222 (85.7%)	223 (87.1%)	445 (86.4%)	
**Feverish**, N (%)				0.832
Grade1	30 (11.6%)	29 (11.3%)	59 (11.5%)	
Grade2	5 (1.9%)	7 (2.7%)	12 (2.3%)	
No	224 (86.5%)	220 (85.9%)	444 (86.2%)	
**Oxygen equipment on day 1**, N (%)				0.281
MV (invasive non/invasive)	5 (1.9%)	11 (4.3%)	16 (3.1%)	
mask reservoir	77 (29.7%)	63 (24.6%)	140 (27.2%)	
oxygen mask/nasal cannula	153 (59.1%)	155 (60.5%)	308 (59.8%)	
none	24 (9.3%)	27 (10.5%)	51 (9.9%)	
**Comorbidities**, N (%)				0.925
0	87 (33.6%)	81 (31.6%)	168 (32.6%)	
1	77 (29.7%)	74 (28.9%)	151 (29.3%)	
2	59 (22.8%)	64 (25.0%)	123 (23.9%)	
>2	36 (13.9%)	37 (14.5%)	73 (14.2%)	
**Macrolides**, N (%)				<0.001
Azithromycin	18 (6.9%)	53 (20.7%)	71 (13.8%)	
Clarithromycin	19 (7.3%)	29 (11.3%)	48 (9.3%)	
None	222 (85.7%)	174 (68.0%)	396 (76.9%)	
**Anticoagulants**, N (%)				0.163
Heparin	16 (6.2%)	9 (3.5%)	25 (4.9%)	
LMWH	229 (88.4%)	237 (92.6%)	466 (90.5%)	
Oral	3 (1.2%)	0 (0.0%)	3 (0.6%)	
None	11 (4.2%)	10 (3.9%)	21 (4.1%)	
**Corticosteroids**, N (%)				0.06
Methylprednisolone	90 (34.7%)	69 (27.0%)	159 (30.9%)	
Prednisone	3 (1.2%)	1 (0.4%)	4 (0.8%)	
Dexamethasone	156 (60.2%)	166 (64.8%)	322 (62.5%)	
None	10 (3.9%)	20 (7.8%)	30 (5.8%)	

N—Total number; IQR—Interquartile range; MV—Mechanical ventilation; O_2_ saturation—Oxygen saturation on admission at room air; LMWH—Low Molecular Weight Heparin. Comorbidities: Hypertension, diabetes, obstructive lung diseases, cardiomyopathies, cancer, obesity, and renal or hepatic impairment.

**Table 2 medicina-59-00934-t002:** Illustrating the difference between the two groups regarding LOS, days of treatment with oxygen, and mortality.

Outcome	Yes, the Colchicine Group (N = 259)	No, the Non-Colchicine Group (N = 256)	Total (N = 515)	*p*-Value
**Hospital length of stay (LOS), Days** Median (IQR)	7.0 (5.0, 9.0)	6.0 (4.0, 8.0)	6.0 (5.0, 9.0)	0.021
**Days of treatment of supplemental oxygen (alive)** Median (IQR)	6.0 (4.0, 9.0)	5.0 (3.0, 7.0)	5.0 (3.0, 8.0)	0.008
**In-hospital mortality** N (%)	49 (18.9%)	55 (21.5%)	104 (20.2%)	0.468

**Table 3 medicina-59-00934-t003:** Summary of included studies (Qualitative analysis).

Quality Assessment	Outcomes	Dose of Colchicine	Competitor	Study Type, Study Period, Setting	Population, Sample Size, Age.	Author, Year
**Randomized trials**						
Poor	Colchicine has less clinical deterioration rate (*p* = 0.02). less event-free survival time (*p* = 0.03).	1.5 mg loading dose followed by 0.5 mg after one hour, then 0.5 mg b.i.d. for 3 weeks.	Colchicine and SOC vs. SOC.	RCT, open-label 3 April to 27 April 2020 Multicenter, Greece.	Hospitalized COVID-19 105 patients Median age 64 (54–76) years	Deftereos, 2020 [21]
Good	Colchicine reduces the length of both, supplemental oxygen therapy (*p*< 0.001) and hospitalization (*p* = 0.003).	0.5 mg t.i.d. for 5 days, followed by 0.5 mg b.i.d. for 5 days.	Colchicine and SOC vs. placebo and SOC	RCT, double-blinded, placebo-controlled. April 2020 to August 2020 Single center, Brazil.	Hospitalized COVID-19, 72 patients Median age 54.5 (42.5–64.5) years.	Lopes, 2021 [14]
Fair	No significant difference in 28-day mortality reduction, hospital length of stay (*p* = 0·44), and risk of mechanical ventilation (*p* = 0·47).	1 mg loading dose, followed by 0.5 b.i.d. for 10 days or until discharge.	Colchicine and SOC vs. SOC.	RCT, open-label. November 2020, and March 2021. Multicenter in UK, Indonesia, and Nepal.	Hospitalized COVID-19, 19,423 patients, Mean age 63·4 ± 13·8 years.	Recovery, 2021 [17]
Good	No significant difference in death, progression or length of stay (*p* = 0.67).	1.5 mg loading dose, followed by 0.5 mg b.i.d. for 10 days	Colchicine vs. placebo.	RCT, triple-blind, placebo-controlled. May 2020 to April 2021. Multicenter, Mexico.	Hospitalized severe COVID-19, 116 patients. Median age 53 (44–62) years.	Absalón-Aguilar, 2021 [22]
Poor	Colchicine reduces the SHOCS-COVID score *, the median SHOCS score decreased from 8 to 2 (*p* = 0.017).	1 mg for 3 days, followed by 0.5 mg/day.	Colchicine vs. ruxolitinib and secukinumab.	RCT, open label. on day 12 or at discharge before day 12 one center, Russian.	Hospitalized later stage COVID-19, 43 patients Mean age 61.9 ± 10.6 years.	Mareev, 2021 [23]
Poor	Colchicine reduces the duration of hospitalization (*p* = 0.001) and the symptoms (fever) (*p* = 0.02).	1 mg o.d. for six days.	Colchicine and SOC vs. SOC.	RCT, open-label, and double-blind. May to June 2020. one hospital, Iran.	Hospitalized COVID-19, 100 patients. Median age 56 years.	Salehzadeh, 2021 [24]
Fair	No significant difference in the reduction of mechanical ventilation, and 28-day mortality (*p* = 0.08).	1.5 mg loading dose followed by 0.5 mg within 2 h of the initial dose and 0.5 mg b.i.d. for 14 days or discharge.	Colchicine and SOC vs. SOC.	RCT, open-label. April 2020 to March 2021. Multi-center, Argentina.	Hospitalized COVID-19, 1279 patients Mean age 61.8 ± 14.6 years.	Diaz, 2021 [25]
Poor	No significant difference in treatment (neither improved the clinical status, nor the inflammatory response) (*p* = 0.303).	1.5 mg loading dose for 2 days, followed by 0.5 mg b.i.d. for one week and 0.5 mg o.d. for 28 days.	Colchicine and SOC vs. SOC. SOC (dexamethasone, remdesivir and tocilizumab or baricitinib)	RCT, open-label Four weeks, Madrid, Spain.	Hospitalized COVID-19 patients (non-ICU) 103 patients Mean age 51 ± 12 years.	Pascual-Figal, 2021 [26]
Poor	Colchicine inhibits the release of α-Def and D-dimer.	1 mg b.i.d. for one-day initial dose followed by 0.5 mg b.i.d. for the other 7 days.	Colchicine and SOC vs. SOC.	Randomized, open-label, controlled, clinical trial Hadassah Hospital, Israel.	Hospitalized COVID-19, 16 patients Mean age 51.4 years	Abdeen, 2021 [27]
Poor	No significant difference in hospital length days or days on oxygen supplementation. Colchicine decreases mortality (21.4%) vs. supportive care (33.3%) vs. budesonide group (35.7%), (*p* = 0.67).	1.5 mg loading dose, followed by 0.5 mg after hour in day 1, then 0.5 mg b.i.d. for 4 days.	Colchicine and supportive care vs. budesonide and supportive care vs. supportive care only.	RCT August 1 to 30. One center, Damascus, Syria.	Hospitalized COVID-19 (non-ventilated). 49 patients. Mean age 50 years.	Alsultan, 2021 [28]
Poor	Colchicine improves clinical status distribution on chest CT evaluation (*p* = 0.048) and reduces pulmonary infiltration (*p* = 0.026).	0.5 mg loading dose for 3 days followed by 1 mg for 12 days.	Colchicine and SOC vs. SOC.	RCT March to September 2020. Five hospitals, Iran.	Hospitalized and outpatients, moderate to severe, COVID-19. 202 patients Median age 56 years	Pourdowlat, 2021 [29]
Fair	No significant difference in hospitalizations, cause of mortality, and need for ventilation with (*p* = 0.96, 0.91, and 0.95), respectively.	0.6 mg b.i.d. for 30 days.	Colchicine and SOC vs. SOC.	RCT May 2020 and March 2021. Multicenter, USA.	Hospitalized COVID-19 with the cardiac disease 93 patients Mean age 71.2 years.	Rabbani, 2022 [30]
Poor	Colchicine decreases the time of recovery by an average of 5 days in severe disease and 2 days in moderate (*p* ≤ 0.001). Did not lower the death rate.	0.5 mg b.i.d. for 7 days, followed by 0.5 mg o.d. for another 7 days.	Colchicine and SOC vs. SOC.	RCT, open-label. April to August 2021. One hospital, Iraq.	Hospitalized moderate and severe COVID-19 160 patients, Median age 49 [37–60.5] years.	Gorial, 2022 [31]
Fair	No significant difference in the combined outcome of (CPAP/BiPAP use, ICU admission, invasive mechanical ventilation, or death) (*p* = 0.533).	1 mg loading dose for 5 days followed by 0.5 mg/day.	Colchicine and SOC vs. placebo and SOC.	RCT, observer-blinded endpoint (PROBE). August 2020–March 2021. Four tertiary university hospitals, Spain.	Hospitalized with COVID-19 without oxygen support. 239 patients, Mean age of 65.1 ± 16.0 years.	Cecconi, 2022 [32]
Poor	Colchicine combination reduces mortality (*p* = 0.009), days of oxygen requirement (*p* = 0.038), and the need for mechanical ventilation (*p* = 0.020)	0.5 mg t.i.d. for 5 days, then 0.5 mg b.i.d. for 14 days, or until discharge	1-Ivermectin + colchicine + SOC. 2-Colchicine + SOC. 3- SOC	RCT. November 2021 to February 2022. University Isolation hospitals, Egypt.	Hospitalized COVID-19, 135 patients. Mean age 57 years.	El Sayed, 2022 [33]
Poor	Colchicine has less musculoskeletal (*p*= 0.001) and respiratory symptoms (*p* = 0.006), high average SpO2 with oxygen (94.05%), (SpO2: 90.46%) with (*p* = 0.029), shorter duration at hospitals days (*p* = 0.009). Less admission to the respiratory care, (*p* = 0.041).	0.5 mg b.i.d. for 14 days.	Colchicine and SOC vs. SOC.	RCT, open label. May to June 2021. One Hospital, Iraq.	Hospitalized or at home COVID-19. 80 patients. Age 18–70 years.	Jalal, 2022 [34]
Poor	No significant difference in oxygen flow, ventilation, or mortality (*p* = 0.58). Additionally, in oxygen flow, ventilation, or respiratory mortality (*p* = 0.58).	1.2 mg loading dose followed by 0.6 mg two hours later and then 0.6 mg t.i.d. for 28 days.	Rivaroxaban, aspirin and SOC vs. colchicine rivaroxaban, aspirin, and SOC.	RCT open-label, 2 × 2 factorial. October 2020, and February 2022 11 countries.	Hospitalized COVID-19, 2749 patients. Mean age 56.1 years.	Eikelboom, 2022 [35]
Fair	No significant difference in the need for mechanical ventilation and death after 14 days (*p* = 0.171). However, after 28 days’ colchicine reduces mechanical ventilation and death (*p* = 0.035).	1.2 mg once initial dose followed by 0.6 mg daily for 13 days.	Colchicine and SOC vs. placebo and SOC.	RCT, blinded placebo controlled. June to November 2020. Dhaka, Bangladesh.	Hospitalized COVID-19, 300 patients Median age 47 (35–55) years.	Rahman, 2022 [36]
Poor	No significant difference in the improvement of severe symptoms including cough, shortness of breath, and oxygen requirement with (*p* = 0.94, 0.69, and 0.28), respectively.	1.5 mg o.d. for two days initial dose followed by 0.5 mg b.i.d. for 6 days, then 0.5 mg o.d. for other 14 days.	Colchicine and SOC vs. SOC.	RCT, open-labeled. December 2020 to July 2021. Abbottabad, Pakistan.	Hospitalized COVID-19, 96 patients Median age 55.0 (47.5, 68.0) years.	Haroon, 2022 [37]
Poor	No significant difference in the day of hospitalization, comorbidities, and oxygen requirements.	Initial dose 0.5 mg t.i.d. for a maximum of 30 days or until hospital discharge.	Colchicine and SOC vs. SOC	RCT open-label April 2020 to May 2021. Multicenter, Italy.	Hospitalized COVID-19 (non-vaccinated). 152 patients Median age of 69.1 ± 13.1 years.	Perricone, 2023 [38]
Poor	No significant difference in improving clinical symptoms and decreasing complications in hospitalized COVID-19 patients (*p* = 0.746).	2 mg once initial dose followed by 0.5 mg b.i.d. for 7 days.	Colchicine and SOC vs. placebo and SOC	RCT, double-blind, placebo-controlled. February to May 2021 One hospital, Iran.	Hospitalized COVID-19, 106 patients Mean age 54.62 years.	Kasiri, 2023 [39]
Poor	Colchicine reduces inflammation and improves symptoms (*p* = 0.018), and reduces the severity score of CT.	0.5 mg o.d.	Colchicine, aspirin, and SOC vs. aspirin and SOC.	RCT, open-label. Two hospitals in Mumbai, India.	Hospitalized moderate COVID-19, 122 patients, Age range 40–80 years.	Sunil Naik, 2023 [40]
Poor	Colchicine inhibits the NLRP3 inflammasome, lowers levels of Casp1p20 and IL-18 in serum (*p* < 0.05). Reduces the supplemented oxygen saturation needed and hospitalization days.	0.5 mg t.i.d. initial dose for 5 days, followed by 0.5 mg b.i.d. for another 5 days.	Colchicine and SOC vs. placebo and SOC	RCT, double-blinded, placebo-controlled. April to August 2020. São Paulo, Brazil.	Hospitalized moderate COVID-19, 72 patients, Median age 55 years.	Amaral, 2023 [41]
**Observational studies**						
Good	Colchicine has a better survival rate (*p* < 0.0001).	1 mg/day	Colchicine and SOC vs. SOC.	One center observational study from March to April 2020. Italy.	Hospitalized COVID-19 262 patients. Mean age 78.4 (7.5) years non-survivors, 66.6 (13.4) years survivors.	Scarsi, 2020 [42]
Good	Colchicine lowers the rate of intubation (*p* < 0.0001), and mortality (*p* = 0.0003), and increases the discharge rate (*p* = 0.0003). Non-significant in mortality and duration of hospitalization in all intubated patients.	0.6 mg b.i.d. for three days, followed by 0.6 mg o.d. for 12 days.	Colchicine and SOC vs. SOC.	Prospective comparative cohort. March to May 202. One hospital, New York City, USA.	Hospitalized COVID-19 182 patients. Mean age 67.7 ± 12.3 years.	Sandhu, 2020 [43]
Poor	Colchicine increases the rate of discharge (*p* = 0.023), and decreases mortality by day 28 (*p* = 0.023).	1.2 mg loading dose for 3 days, followed by 0.6 mg b.i.d.	Colchicine and SOC vs. SOC.	Single-center propensity score matched 1:1 cohort study, March to May 2020. Community Teaching Hospital, USA.	Hospitalized severe COVID-19 66 patients Mean age 61.2 ± 13.0 years.	Brunetti, 2020 [44]
Poor	Colchicine combination reduces mortality rate (*p* < 0.05). No significant differences in clinical severity between the groups (*p* > 0.05).	For 20 days.	1-Broad-spectrum antibiotics + low molecular weight heparin (LMWH) + corticosteroids + colchicine. 2-Antibiotic + LMWH + corticosteroids. 3-LMWH + corticosteroids. 4-LMWH + corticosteroids + colchicine. 5-Other treatments (Tocilizumab).	Descriptive observational study. May to August 2020, Private third-level clinic, Colombia.	Hospitalized COVID-19 209 patients. Median age 60 years.	García-Posada, 2021 [45]
Good	Colchicine reduces 21-day mortality (*p* = 0.006), and accelerates recovery.	1 mg o.d. from 1–21 days or until clinical improvement.	Colchicine and SOC vs. SOC.	Retrospective cohort study, February to April 2020. A tertiary health-care Centre in Parma, Italy.	Hospitalized with COVID-19 with pneumonia, 141 patients Mean age 60.5 (13.4).	Manenti, 2021 [46]
Good	Colchicine lowers mortality (*p* = 0.179).	0.5 mg b.i.d. for 7 to 14 days.	SOC + colchicine + corticosteroids vs. SOC + corticosteroids vs. SOC alone.	Cross-sectional study March to August 2020. Three clinics in Antioquia, Colombia.	Hospitalized COVID-19, 301 patients. Mean age 56.8 (±17.3) years.	Pinzón, 2021 [47]
Good	Colchicine reduces the length of hospital stays (*p* < 0.001). No significant difference in ICU admission, anti-inflammatory administration, or mortality. Only colchicine (1 mg/day dose) reduces mortality (*p* = 0.031) and ICU admission rate (*p* = 0.011) compared with 0.5 mg/day dose.	0.5 mg/day and 1 mg/day In two separate group	Colchicine and SOC vs. SOC.	Retrospective cohort. August to December 2020. One hospital, Turkey.	Hospitalized COVID-19 336 patients Mean age 62.72 ± 14.37 years.	Karakaş, 2022 [48]
Good	No significant difference in decreasing progression concerning admission to the intensive care unit, mortality rate, and treatment failure with (*p* = 0.174, 1.000, and 0.505), respectively.	0.5 mg b.i.d.	Colchicine and SOC vs. SOC.	Retrospective case-control study. October 2020 to October 2021. Two centers, Turkey.	Hospitalized COVID-19. 330 patients. Mean age 59.37 ± 14.78 years.	Doğan, 2022 [49]
Good	Colchicine resolves symptoms and decreases in duration of hospital stay (*p* < 0.001), ICU admission (*p* = 0.013) need for invasive mechanical ventilation (*p* = 0.025), need for noninvasive mechanical and duration of ICU stay (*p* > 0.05), and lower mortality rate (*p* = 0.006).	1.5 mg initial dose for one day, followed by 0.5 mg b.i.d. on days 2–7 and continuing with 0.5 mg o.d. until completing 14 days.	Colchicine and SOC vs. SOC.	Retrospective study. January to May 2021. One hospital, Egypt.	Hospitalized COVID-19 100 patients. Mean age 58.03 ± 10.59 years.	Korra, 2022 [50]
Abstract	Colchicine reduces the duration of hospitalization (*p* = 0.18). No significant difference in the prevention of ARDS or 28 days mortality.	Treated with colchicine before, or during hospitalization.	Colchicine and SOC vs. SOC.	Retrospective analysis, identified using the Society of Critical Care Medicine COVID-19 registry VIRUS, US.	Hospitalized COVID-19, 108 patients.	Nazir, 2022 [51]
Good	Colchicine decreases ICU admission (*p* = 0.004) and oxygen demand (*p* = 0.01). The adjusted hazard ratio for hospital death is 0.35, (*p* < 0.0001).	0.5 mg b.i.d. within 48 h of declined oxygen saturation.	Colchicine and SOC vs. SOC.	Retrospective, single-center cohort study. November 2020 to January 2021. One hospital, Egypt.	Hospitalized severe COVID-19 153 patients. Mean age 62.65 ± 11.14 years.	Qenawy, 2022 [52]
Good	Colchicine lowers the risk of death (*p* = 0.031).	The median dose was 7.5 mg (3.5–12).	Colchicine and SOC vs. SOC.	Retrospective, multi-center, cohort study. March to June 2020. Two hospitals in Madrid.	Hospitalized COVID-19 (non-ICU). 222 patients. Median age 79 (66–88) years.	Villamañán, 2022 [53]
Good	Colchicine does not affect modifying the risk of hospitalization (*p* = 0.678), preventing (*p* = 0.291), or decreasing the severity (*p* = 0.889).	Not mentioned	Colchicine alone or in combination.	Retrospective, case-control, cohort study. June to December 2020, Spain.	Hospitalized COVID-19 86,602 patients (3060 hospitalized, 26 757 not hospitalized for COVID-19, and 56 785 healthy controls) Median age 74 (59−84).	Sáenz-Aldea, 2023 [54]

* The SHOCS-COVID score includes the assessment of the patient’s clinical condition, CT score of the lung tissue damage, the severity of systemic inflammation (CRP changes), and the risk of thrombotic complications (D-dimer). RCT—randomized controlled trial; SOC—standard of care; o.d—once daily; b.i.d.—two times a day; t.i.d.—three times a day; ACT—Anti Coronavirus Therapies trials; ARDS—acute respiratory distress syndrome.

## Data Availability

Data will be available upon request from the first or corresponding authors.

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
