# Peer review of "Oxygen Saturation in Hospitalized COVID-19 Patients and Its Relation to Colchicine Treatment: A Retrospective Cohort Study with an Updated Systematic Review"

_medicina, 2023, doi:10.3390/medicina59050934_

Round 1

Reviewer 1 Report

Sharaf and colleagues reported the effect of using Colchicine as cytokine storm blocking agents. Poorer outcomes were reported in groups receiving Colchicine, therefore suggesting that this kind of management should not be used. In addition, a systematic review was performed to strengthen the findings. However, I have several concerns that need to be addressed by authors, they are:

1.     Abstract has been presented in a clear and concise manner.

[Introduction]

2.     Line 51. “to date” please state when exactly the data were retrieved.

3.     The absence of safe and efficacious antiviral, despite the continuous research, should be highlighted as the challenge of COVID-19 management. Please refer to:

a.     https://doi.org/10.52225/narra.v3i1.98

b.     https://doi.org/10.52225/narra.v2i3.92

c.     https://doi.org/10.3390/scipharm91010015

4.     Line 66. “the optimal time” consider “crucial time”. The optimal time would be during the mild/moderate COVID-19, before progressed to the later stage as current antivirals work during the stated phase.

5.     Line 72. Add full stop (.) after “…[7]”

6.     Line 81. “…severe complications…” such as?

[Methods]

7.     In the study design, please state the date and the total number of patients.

8.     Please remove (:) form the sub-section titles.

9.     In inclusion criteria, please explain about the severity of COVID-19 and how it was determined.

10.  Did authors exclude those with immunocompromised conditions? What about those in pregnancy and lactating? Also, what about those in malignancy? These groups might have different immune responses.

11.  All lab findings such as ALT, AST, CRP, etc., how they were determined (using what analytical methods/tools) should be explicitly stated.

12.  “2.5.1 Database search..” I think a table explaining about who perform the search on which database should be presented. The table could also explain about the keywords combination used.

13.  Please state clearly how the data are managed. For instance, did authors convert the median into mean? All data conversion should be stated.

14.  I encourage authors to apply critical appraisal to the included studies.

Results

15.  Table 1. “Placebo group” did author perform the blinding? Please state clearly.

16.  Heart rate, respiratory rate, (and others!!) should be presented with their respective units!

17.  I think the adjustment with confounder should also be performed for the respiratory rate.

18.  It is better to simplify Table 3.

Discussion

19.  Discussion has been performed in comprehensive manner. But maybe dividing the section into three subsections (Finding from the cohort study; Findings from systematic review; and strength and limitation) would improve the readability.

Conclusions

20.  “substantial size” is this based on objective criteria or rather a subjective parameter?

21.  Why authors recommend further RCT if they found that the colchicine is associated with poorer outcome?

The writing quality is sufficient for publication; it is easy to follow - but several technical issues persist (see comments above).

Author Response

Reviewer 1: Comments and Suggestions for Authors

Sharaf and colleagues reported the effect of using Colchicine as cytokine storm blocking agents. Poorer outcomes were reported in groups receiving Colchicine, therefore suggesting that this kind of management should not be used. In addition, a systematic review was performed to strengthen the findings. However, I have several concerns that need to be addressed by authors, they are:

Reply: we appreciate your helpful evaluation and suggestion to increase our manuscript impact.

  1. Abstract has been presented in a clear and concise manner.

[Introduction]

  1. Line 51. “to date” please state when exactly the data were retrieved.

Reply: Thank you for this suggestion, we added “till March 2023, 761.4 million cases and 6.9 million fatalities have been reported.”

  1. The absence of safe and efficacious antiviral, despite the continuous research, should be highlighted as the challenge of COVID-19 management. Please refer to:
  2. https://doi.org/10.52225/narra.v3i1.98
  3. https://doi.org/10.52225/narra.v2i3.92
  4. https://doi.org/10.3390/scipharm91010015

Reply: Thank you for this suggestion, we added “The molecular pathophysiology and mechanism behind COVID-19 infection is currently incomplete and unclear. Thus, unanticipated consequences might be disregarded in the rush to create COVID-19 treatment options [59]” within the discussion section

  1. Line 66. “the optimal time” consider “crucial time”. The optimal time would be during the mild/moderate COVID-19, before progressed to the later stage as current antivirals work during the stated phase.

 Reply: Thank you for this explanation, and we changed the optimal time with the crucial time at line 70.

  1. Line 72. Add full stop (.) after “…[7]”

Reply: Thank you for this comment, Done.

  1. Line 81. “…severe complications…” such as?

Reply: Thank you for this suggestion, we added: “This feature helps prevent the complications mentioned earlier and acts as a cytokine storm-blocking agent [10].”

[Methods]

  1. In the study design, please state the date and the total number of patients.

Reply: Thank you for this important suggestion, we added “by using a deep nasopharyngeal swab from the 1st of June until the end of November 2021”, line 122,  and a paragraph of sample size calculation line [161-165]

  1. Please remove (:) from the sub-section titles.

Reply: Done, thanks.

  1. In inclusion criteria, please explain about the severity of COVID-19 and how it was determined.

Reply: Thank you for this constructive comment, we added (wards, intermediate care units, or intensive care units) were diagnosed with COVID-19 infection with any degree of infection through polymerase chain reaction (PCR), with any degree of severity according to the Egyptian MoHP protocol version 4, September 2021 [18], Figure S2, supplementary.

  1. Did authors exclude those with immunocompromised conditions? What about those in pregnancy and lactating? Also, what about those in malignancy? These groups might have different immune responses.

Reply: Thank you for this query, but we did not exclude any of those patients (we included all patients who receive treatment in the selected hospitals and time period), but we did not meet pregnancy or lactating women.

  1. All lab findings such as ALT, AST, CRP, etc., how they were determined (using what analytical methods/tools) should be explicitly stated.

Reply: We added “complete blood count (CBC) by flowcytometry, C-reactive protein (CRP) by latex agglutination, random blood sugar (RBS) by glucose oxidase/hydrogen peroxide, urea by urease/UV, serum creatinine (sCr) by alkaline picrate, alanine aminotransferase (ALT), and aspartate aminotransferase (AST) by UV without P5P” [ line 136 -139].

  1. “2.5.1 Database search..” I think a table explaining about who perform the search on which database should be presented. The table could also explain about the keywords combination used.

Reply: Thank you very much for this important suggestion, the performed database search was described by initials in paragraph [165 -175}, then we created a supplementary file containing detailed search strategy table S1.

  1. Please state clearly how the data are managed. For instance, did authors convert the median into mean? All data conversion should be stated.

Reply: Thank you for this comment, in line 151 “First descriptive analysis and appropriate tests based on normality”. Therefore, we reported continuous data according to the Shapiro test of normality, mean and Sd for normally distributed data, and median for not normally distributed data.

  1. I encourage authors to apply critical appraisal to the included studies.

Reply: Thank you very much for this important suggestion, we performed a quality assessment of all included studies. And added a paragraph describing the studies  line [243 -254]

Results

  1. Table 1. “Placebo group” did author perform the blinding? Please state clearly.

Reply: Thank you for this comment, but It is a retrospective, collected patients’ data from hospital records, and all patients have received standard care of COVID-19 treatment.

from the 1st of June until the end of November 2021, either administering colchicine tablets or not according to the Egyptian MoHP protocol version 4, September 2021??? Should state two protocols??

 Reply: We added the different treatment protocols to the supplementary file table S3.

  1. Heart rate, respiratory rate, (and others!!) should be presented with their respective units!

Reply: Thank you very much for this comment, we added all units to Table 1.

  1. I think the adjustment with confounder should also be performed for the respiratory rate.

Reply: Thank you for this comment, we noticed that RR was a highly statistically significant difference, but clinically has no meaning and does not affect the severity, therefore we didn’t adjust for this parameter.

  1. It is better to simplify Table 3.

Reply: Thank you for this suggestion, done.

Discussion

  1. Discussion has been performed in comprehensive manner. But maybe dividing the section into three subsections (Finding from the cohort study; Findings from systematic review; and strength and limitation) would improve the readability.

Reply: Thank you very much for this important suggestion, Done.

Conclusions

  1. “substantial size” is this based on objective criteria or rather a subjective parameter?

Reply: Based on objective criteria (large sample size regarding other observational studies)

  1. Why authors recommend further RCT if they found that the colchicine is associated with poorer outcome?

Reply: Thank you for this comment, we deleted this statement.

This study is an observational study so the causal relationship between adverse events and colchicine administration could not be well defined.

Comments on the Quality of English Language

The writing quality is sufficient for publication; it is easy to follow - but several technical issues persist (see comments above).

Reviewer 2 Report

Please

1. make the table the same name of the groups as in the text. In the results you wrote about colchicine and non-colchicine groups in the table you have colchicine and placebo groups

2. give more statistical tests in part 2.4. statistical analysis - how you compared age? etc.

3. Do you have statistical changes in the pharmacotherapy between groups? Why "does the placebo" group have more macrolides and corticosteroids?

4. In Table 2 you have yes and no - is it about colchicine treatment?

Author Response

Reviewer 2: Comments and Suggestions for Authors

Please

  1. make the table the same name of the groups as in the text. In the results you wrote about colchicine and non-colchicine groups in the table you have colchicine and placebo groups

Reply: Thank you for this important comment, we categorized Tables 1, and 2 as the Colchicine group (N=259, and the non-Colchicine group (N=256)

  1. give more statistical tests in part 2.4. statistical analysis - how you compared age? etc.

Reply: Thank you for this comment, we added ), tests of significance between 2 groups were either a t-test or Mann-Whitney, line 153.

  1. Do you have statistical changes in the pharmacotherapy between groups? Why "does the placebo" group have more macrolides and corticosteroids?

Reply: Thank you for this observation, although we have statistically significant differences in macrolides and corticosteroids, we added them to the Cox-hazard regression model to adjust for these confounders.

  1. In Table 2 you have yes and no - is it about colchicine treatment?

Reply: Thank you for this suggestion, done Yes = colchicine treatment group, No = non-colchicine treatment group

Round 2

Reviewer 1 Report

Thank you for the revision. There are some errors left, please improve them.

1. "till March 2023" consider "as of [date] March 2023"

2. revised to "....serum creatinine (sCr) by alkaline picrate, and alanine aminotransferase (ALT) and aspartate aminotransferase (AST) by UV without P5P." Also, what UV? UV-vis spectrometry? Also, add the wavelength; 340 nm?

3. Table 3. Descriptions of "Outcomes" are too lengthy. Make point-by-point.

4. About the critical appraisal. First, please move all protocol statements to methods. Second, please move the paragraph to portrait orientation. 

5. "To the best of our knowledge,"; "Two RCTs have same our primary end point"; and others.. why did you bold them?? I meant discussion should be divided using sub-section title, not make the first sentence of the paragraph with bold.

6. In conclusion, I am still confused why authors suggest to perform further RCT; I mean, is it still worth it to investigate Colchicine treatment for COVID-19 patient?

Authors should used more formal tone in the writing.

Author Response

Comments and Suggestions for Authors
-Round 2

Thank you for the revision. There are some errors left, please improve them.

  1. "till March 2023" consider "as of [date] March 2023"

Reply: precise date was added.

  1. revised to "....serum creatinine (sCr) by alkaline picrate, and alanine aminotransferase (ALT) and aspartate aminotransferase (AST) by UV without P5P." Also, what UV? UV-vis spectrometry? Also, add the wavelength; 340 nm?

Reply: the corrections were made.

  1. Table 3. Descriptions of "Outcomes" are too lengthy. Make point-by-point.

Reply: Thank you for your comment, it is reduced to the extent that does not prejudice the explanation

  1. About the critical appraisal. First, please move all protocol statements to methods. Second, please move the paragraph to portrait orientation. 

Reply: The correction required was done.

  1. "To the best of our knowledge,"; "Two RCTs have same our primary end point"; and others.. why did you bold them?? I meant discussion should be divided using sub-section title, not make the first sentence of the paragraph with bold.

Reply: bold was removed and the sub-titles were added with justified according the journal style (italic).

  1. In conclusion, I am still confused why authors suggest to perform further RCT; I mean, is it still worth it to investigate Colchicine treatment for COVID-19 patient?

Reply: The RCT sentence was removed.

Comments on the Quality of English Language

Authors should used more formal tone in the writing.

English language was revised and corrected.
